# In Vivo Optical Imaging of Bladder Cancer Tissues in an MB49 Bladder Cancer Orthotopic Mouse Model Using the Intravesical or Intravenous Administration of Near-Infrared Fluorescence Probe

**DOI:** 10.3390/ijms24032349

**Published:** 2023-01-25

**Authors:** Katsunori Teranishi

**Affiliations:** Graduate School of Bioresources, Mie University, 1577 Kurimamachiya, Tsu 514-8507, Japan; teranisi@bio.mie-u.ac.jp; Tel.: +81-59-231-9615

**Keywords:** ASP5354, bladder cancer, diagnosis, fluorescence imaging, near-infrared fluorescence

## Abstract

Bladder cancer was the twelfth most common cancer worldwide in 2020. Although bladder cancer has been diagnosed using macroscopic techniques, such as white-light cystoscopy and fluorescence blue-light cystoscopy, there is a need to explore more effective noninvasive optical imaging techniques for accurate bladder cancer diagnosis. This study demonstrates the high effectiveness of the near-infrared fluorescence (NIRF) probe ASP5354, which has been developed for ureteral identification during in vivo diagnosis of bladder cancer in an MB49 bladder cancer orthotopic mouse model. After the intravesical injection of 2.4 μM ASP5354 followed by bladder rinsing with saline at 5 min post injection or intravenous administration of ASP5354 at 240 nmol/kg mouse body weight, followed by a waiting period of 5–24 h in mice, ASP5354 was absorbed specifically by cancerous tissue and not by normal tissues in the bladder. NIRF of ASP5354 in cancer tissues was detected using the NIRF imaging camera system. The NIRF clearly showed a boundary between cancerous and normal tissues. Therefore, ASP5354 provides noninvasive and specific optical in vivo imaging of MB49 bladder cancer using intravesical or intravenous injection of ASP5354. ASP5354 may allow for new diagnostic applications for bladder cancer in humans.

## 1. Introduction

In 2020, over 573,000 cases of bladder cancer were diagnosed worldwide, making it the twelfth most common type of cancer worldwide. In addition, the number of male patients with bladder cancer was 3.3 times higher than that of female patients, making bladder cancer the sixth leading cancer type in males [1]. In the same year, nearly 212,536 bladder cancer-related deaths were reported worldwide. Most bladder cancer patients require continuous surveillance and treatment. Nearly 70% of diagnosed cases of bladder cancer are non-muscle-invasive bladder cancer (NMIBC) at stage carcinoma in situ (CIS), Ta, or T1. Early-stage bladder cancer is usually treated with transurethral resection, followed by cystoscopy through the urethra.

Approximately 30% of NMIBC cases progress to muscle-invasive bladder cancer (MIBC) at stages > T1. Jordan and Meeks proposed two potential mechanisms underlying the progression of bladder cancer from stage T1 to MIBC [2]. While one hypothesis states that resected T1 cancer could be understaged and actually be T2 cancer, an alternative hypothesis is that the recurrence of predominantly aggressive cancer could lead to progression to muscle invasion. MIBC requires radical cystectomy, followed by urinary tract reconstruction [3]. It has been reported that radical cystectomy could be related to complications, thereby resulting in a negative impact on patients’ quality of life [4]. Hence, the early and reliable detection and treatment of bladder cancer are of utmost significance.

Most bladder cancer cases are detected by diagnostic testing promoted by hematuria. Hematuria also indicates the presence of diseases related to cystitis or ureteral stones. Techniques such as white-light cystoscopy (WLC), ultrasound, computed tomography urography, and magnetic resonance imaging (MRI) are used to diagnose bladder cancer. Except for WLC, all other techniques are unable to detect subtle urothelial changes and small structural bladder tumors. Hence, WLC has been considered the gold standard for the diagnosis of bladder tumors for several years [5,6]. However, WLC also has some limitations, including its inability to grade cancer stage, determine the infiltration status, and detect early flat cancer at stage CIS [6,7,8].

Kriegmair et al. reported fluorescence cystoscopy following intravesical administration of 5-aminolevulinic acid (5-ALA) to detect urothelial neoplasia [9]. This macroscopic technique has also been established for detecting bladder cancer [blue-light cystoscopy (BLC)] and has been used as an adjunct to WLC over the last 20 years [5,6,7,8]. In BLC diagnosis, after intravesical administration of 5-ALA or its derivative hexyl aminolevulinic acid, the bladder mucosa is exposed to blue light (380–480 nm), which helps in the visualization of cancer cells through red fluorescence emission. In addition, narrow-band imaging (NBI), using blue and green light, has been used as a macroscopic technique to visualize small lesions in the bladder epithelium without the need for the administration of a contrast probe [6,10]. Although BLC and NBI can detect bladder cancer at an early stage, including CIS, these methods cannot determine the infiltration status of cancerous cells because of the low permeability of visible light to and from the bladder tissue.

Owing to its noninvasiveness and real-time imaging, visualization of tumors using optical imaging techniques (such as BLC and NBI) is advantageous for diagnosing early-stage bladder cancer and symptomatic tumors. Key et al. demonstrated multimodal imaging of bladder cancer using nanoparticles, enabling both MRI and near-infrared fluorescence (NIRF) imaging in mice implanted with K9TCC canine bladder cancer cells in the flank [11]. Yuan et al. suggested that the fluorescent QD605-PSCA, which comprises fluorescent quantum dots (*λ*_max_ emission, 605 nm) and prostate stem cell antigen monoclonal antibody, is advantageous for the targeted imaging of EJ human bladder urothelial cancer cells; however, these results have not been validated in vivo [12]. Huang et al. reported in vivo fluorescence imaging of bladder cancer using the probe CyP1 in a murine orthotopic bladder tumor syngeneic model [13]. In their study, CyP1 emitted low fluorescence and was efficiently transported to the bladder through effective renal clearance following intravenous administration, and aminopeptidase N overexpression on bladder cancer transferred CyP1 to near-infrared fluorescent compound CCD, resulting in augmented fluorescence emission (*λ*_max_, 710 nm) in the cancerous bladder. However, they did not present the degree of CCD uptake by cancer cells or imaging of cancer distribution in the mouse bladder [13].

These probes have, at present, not been assessed for safety and ability to clearly visualize the distribution of the width and depth of bladder cancer, even in animals. In 2020, an NIRF imaging probe, CD-NIR-1, was reported for intraoperative ureteral identification and diagnosis [14]. CD-NIR-1 emitted fluorescence at *λ*_max_ 801 nm in phosphate-buffered saline (PBS, pH 7.4), with a fluorescence quantum yield of 0.19. As NIRF (700–900 nm) allows deep tissue penetration of light, low autofluorescence from tissue, and low light absorption and scattering by tissue, this optical feature is suitable for in vivo optical imaging of living tissues. In addition, CD-NIR-1 was transferred into the urine through specific and ultra-rapid renal clearance with no chemical modification, following the intravenous administration of CD-NIR-1 in rat, mouse, and macaque models. In 2021, an analog of CD-NIR-1, ASP5354 (Figure 1), formerly termed TK-1, which emits fluorescence at *λ*_max_ 815 nm in human venous whole blood [15], was preclinically demonstrated to be selectively excreted via the kidneys to the bladder, and indicated no significant toxicity in a toxicological study in cynomolgus monkeys [16], and assessed the safety/tolerability and pharmacokinetics in its first-in-human phase 1 [17]. In this study, real-time imaging of bladder cancer using ASP5354 was investigated in a well-characterized mouse model in which MB49 mouse bladder cancer cells were orthotopically and syngeneically implanted in the mouse bladder [18].

## 2. Results

### 2.1. In Vitro NIRF Imaging of Cellar Uptake of ASP5354

The cellular uptake of ASP5354 in MB49 mouse bladder cancer cells was assessed by incubating the cells in a 24-μM solution of ASP5354, followed by NIRF imaging using a NIRF microscope (Figure 2). NIRF signals of ASP5354 with a 60-s exposure time after a 10-min incubation period were obtained.

### 2.2. NIRF Imaging of Normal Bladder via Intravesical Administration of ASP5354

After 2.4 μM ASP5354 was injected into the normal bladder, incubated for 5 min, and removed using saline; no NIFR signal was observed at an excitation level of 10 a.u. (highest sensitivity) (Figure 3). In the case of incubation with 24-μM ASP5354 for 5 min, NIRF was observed in normal bladder tissues even at low excitation levels (Figure 3).

### 2.3. NIRF Imaging of Bladder Cancer via Intravesical Administration of ASP5354

In vivo NIRF imaging of bladder cancer was performed after intravesical administration of 2.4 μM ASP5354, 1- or 5-min incubation, and subsequent rinsing of the bladder inside with saline five times. Incubation for 1 min showed a slight NIRF signal; however, 5-min incubation demonstrated significant NIRF intensity (Figure 4). These results and the results mentioned in Section 2.2 indicate that a 5-min incubation of 2.4 μM ASP5354 in the bladder is suitable for NIRF imaging of bladder cancer. The NIRF intensity in cancer tissues was dependent on the degree of cancer by visual diagnosis based on the red lesion and its width under whight light (Figure 5 Outside, Inside, and Section). The cancer tissues at a severe degree emitted higher NIRF intensity than cancer tissues at a mild degree in imaging from the outside and inside of the bladder and in the bladder section. No NIRF was emitted in severe blood clots, as shown in photos for severe cancer in Figure 5; however, because the bladder clot is easily visible, the NIRF technique may be unnecessary. NIRF images obtained from outside the bladders were unclear because of light scattering by saline and the bladder layer; however, NIRF images obtained inside the opened bladders clearly showed boundaries between cancer and normal tissues (Figure 5 and Figure 6). As shown in the sections in Figure 5 and Figure 7, ASP5354 at 2.4 μM concentration penetrated deep into cancer tissue in 5 min, and significantly emitted NIRF in the cancer tissues. When bladder cancer tissues were on the back side of the bladder and NIRF images were obtained from the ventral side, NIRF images of the cancer tissues were unclear due to light scattering. In this case, NIRF imaging from the back side of the bladder allowed clear imaging (Figure 8). After completing NIRF imaging, the bladder was rinsed 5 times (total 10 times) and 10 times (total 15 times) with saline; the NIRF intensity decreased compared with that before rinsing at the same excitation level (Figure 9). However, the NIRF intensity was reversed when the excitation level was increased.

### 2.4. NIRF Imaging of Transition of ASP5354 into the Bladder after Intravenous Injection

The transition of ASP5354 into the bladder after intravenous administration of a single dose of 240-nmol/kg body weight was measured using a PDE-neo camera system (Figure 10). The NIRF signal of ASP5354 was observed in the bladder at 10-min post-administration of ASP5354 and in the bladder increased at 60-min post-administration. After 5 h, NIRF signal was observed, and the NIRF was deleted by rinsing with saline five times. At 16 and 24 h post-administration, no NIRF signal was detected in the bladders, even at the highest excitation level.

### 2.5. NIRF Imaging of Bladder Cancer via Intravenous Administration of ASP5354

In vivo and ex vivo NIRF imaging of bladder cancer using ASP5354 was performed from the outside and inside of the bladder at 5, 16, and 24 h after intravenous administration of a single dose of 240 nmol/kg body weight (Figure 11). In the case of imaging at 5 h post-administration, the bladder inside was rinsed with saline five times because of the remaining free ASP5354, while bladder rinsing with saline was not required for imaging at 16 and 24 h following administration. In the imaging of bladders 5, 16, and 24 h post-administration, the normal bladder emitted no NIRF signal (no photos are shown in Figure), and significant NIRF signals were observed in cancer tissues in the cancerous bladder. These results indicate that intravenous administration of ASP5354 can noninvasively image MB49 bladder cancer cells 5–24 h post-administration.

## 3. Discussion

This study evaluated the ability of ASP5354 to perform in vivo and ex vivo NIRF imaging of bladder cancer. The MB49 mouse bladder cancer model was used in this study, which has been extensively used for >40 years to examine imaging with ASP5354 [18,19].

A recently published study reported the development of near-infrared fluorescent probes with specific renal clearance after intravenous administration for intraoperative ureteral identification and diagnosis to avoid intraoperative iatrogenic ureteral injuries [14]. Subsequently, ASP5354 was successfully developed and preclinically studied for clinical applications [16]. In the present study, it was demonstrated that ASP5354 aids in the clear visualization of bladder cancer, differentiated from normal bladder tissues, through intravesical or intravenous administration, and that real-time imaging techniques using ASP5354 and NIRF imaging devices could be potentially used to diagnose bladder cancer in the clinical setting.

Optical imaging of a living body requires deep tissue penetration, low autofluorescence, low light absorption, and low light scattering by the tissues [20]. Therefore, the use of near-infrared light is a potentially robust strategy for the detection of lesions and invisible tissues and image-guided surgery. Although several near-infrared fluorescent probes for ureteral visualization through intravenous administration have been reported based on the high capacity of renal excretion, these probes have not been applied to real-time imaging of bladder cancer [21,22,23,24,25,26,27,28,29]. An NIRF diagnostic, indocyanine green (ICG), is clinically available and has recently been used for imaging-guided surgery such as fundus angiography, identification of sentinel lymph nodes, and perfusion imaging [30]. ICG specifically accumulates in the liver following intravenous administration and facilitates the identification of liver cancer [31,32]. In addition, negligible transfer of ICG into the kidneys could occur, resulting in its use in imaging kidney cancer [33,34]. Nevertheless, evaluation of NIRF imaging of bladder cancer using ICG has not yet been reported.

In this study, experiments for imaging the cellular uptake of ASP5354 in MB49 cells revealed that MB49 cells could effectively uptake ASP5354. In addition, after ASP5354 was intravesically administered at a single dose of 2.4 μM, and 5 min after that nonadsorbing ASP5354 was removed by rinsing five times with saline, NIRF of ASP5354 in cancer tissues was observed from outside of bladders using a clinically commercially available NIRF imaging device, whereas no NIRF of ASP5354 was emitted from normal tissue in the bladder. Additional rinsing decreased the NIRF intensity; however, the NIRF intensity recovered by increasing the excitation level of the camera system. These findings revealed that ASP5354 can be used to clearly and reliably visualize cancer tissues using the NIRF imaging technique. NIRF imaging was performed from outside the bladder, indicating that this imaging technique could be used even in open or laparoscopic bladder surgery. However, observation from many directions is required to obtain clear images because light scattering must be reduced. Notably, since NIRF cystoscopy for the mouse bladder could not be used in this study, no in vivo NIRF images from the inside of the bladder were acquired. However, when observed from the inside of the bladder after opening, the cancer distribution could be clearly viewed as NIRF images.

In addition, ASP5354 was transferred to the bladder through renal filtration following a single intravenous administration of 240 nmol/kg mouse body weight. The cancer tissues absorbed ASP5354, and later urine (with low or no probe concentration) washed away the non-adsorbing ASP5354. Subsequently, 5, 16, and 24 h after intravenous administration, NIRF was successfully observed in cancer tissues, but not in normal tissues. The proposed imaging method should potentially shorten the diagnosis time for bladder cancer in the clinical setting compared with the intravesical administration of ASP5354, since the latter requires a waiting time of approximately 5 min for uptake into cancerous tissue.

Studies have shown that while the cancer tissues in the liver emit NIRF of ICG after a few days of intravenous administration of ICG, the healthy tissues do not show any fluorescence emission [31,32]. In addition, ICG can be absorbed by differentiated hepatocellular carcinoma cells and retained in the cytoplasm or pseudo-glands for several weeks after intravenous administration [35]. Conversely, a study reported that kidney cancer emitted no NIRF during ICG administration, unlike normal tissues [33,34]; however, the reason for this observation has not yet been elucidated. The present study demonstrated the specific uptake of ASP5354 in MB49 cancer tissues. Key et al. demonstrated the utility of near-infrared fluorescent peptide-conjugated glycol chitosan nanoparticles, which are based on the enhanced permeability and retention (EPR) effect of nanoparticles for tumors and targeting peptides for in vivo imaging of bladder tumors [11,36]. In addition, fluorescent probes such as QD605-PSCAF (having a prostate stem cell antigen monoclonal antibody), CyP1 (having a substrate for aminopeptidase N overexpressed in the cancerous bladder), and anti-CD47 (having an antibody for CD47 as a surface marker of human solid tumors) were designed according to targeted molecular imaging [12,13,37]. It is remarkable that the structurally simple and small-sized molecule ASP5354 (molecular weight: 3079) has no targeted molecular structure and no macromolecular EPR effect, but has the ability to be selectively uptaken in bladder cancer tissues. This new function of ASP5354 could be utilized for imaging other cancers in the future.

This study had a few limitations that need to be acknowledged. Although ASP5354 can be used with NIRF camera devices for real-time imaging of bladder cancer, to the best of my knowledge, currently there are not any NIRF cystoscopy systems commercially available for use in animals and humans. Additionally, the NIRF cystoscopy system requires a source for excitation light at approximately 780 nm, a detector for fluorescence (> 800 nm), and optical filters, unlike the BLC system in which blue light for excitation is irradiated and red fluorescence is detected. Nevertheless, considering that NIRF endoscopic imaging and NIRF laparoscopic imaging are currently used for surgery, fusion of these NIRF imaging technologies and BLC technology can allow the development of new NIRF cystoscopy systems, and a combination of the NIRF cystoscopy system and ASP5354 can provide an efficient method for bladder cancer diagnosis in clinical settings. The normal thickness of the human bladder wall is approximately 1.5 cm and it becomes thinner as urine accumulates. The in vivo mouse bladder wall is thinner (<0.1 mm) than human bladder wall. In this study using mouse model, the cancer invasion into the deep bladder layer was uncontrolled after implantation due to thinness of bladder wall. Then, no experiments to evaluate difference in in vivo NIRF imaging between non-muscle-invasive bladder cancer and muscle-invasive bladder cancer using ASP5354 was successfully performed. The results from this study demonstrated the efficacy of ASP5354 in visualizing MB49 bladder cancer in a mouse model, while the concentration of ASP5354, incubation time for intravesical injection method, dose of ASP5354, and imaging timing of intravenous administration method for bladder cancer patients should be optimized according to the thickness of the bladder layer and the NIRF imaging devices used in clinical settings, and the efficacy and safety of this technique in patients with many forms and stages of bladder cancer need to be established. Nonetheless, the findings of this study can act as a first step toward achieving these goals.

This study demonstrated that the NIR fluorescent probe ASP5354 can aid in the imaging of bladder cancer based on NIRF emission via specific and robust absorption in cancer tissues. ASP5354 can visualize bladder cancer within 24 h after a single-dose intravenous administration or 5 min after a single-dose intravesical administration using the NIRF camera system. The chemical structure of ASP5354 is simple and ASP5354 can be easily prepared at low cost.

## 4. Materials and Methods

### 4.1. Materials

In this study, ASP5354 (C_135_H_197_N_4_O_73_Cl, molecular weight: 3079), formerly termed TK-1, was prepared as described [15]. Dulbecco’s modified Eagle’s medium (DMEM) and penicillin–streptomycin solution (5000 U/mL) were purchased from Thermo Fisher Scientific (Tokyo, Japan). Fetal bovine serum (FBS) was purchased from MP Bio Japan (Tokyo, Japan), ketamine from Daiichi Sankyo Propharma Co., Ltd. (Tokyo, Japan), medetomidine and atipamezole from Kyoritsu Seiyaku Co., Ltd. (Tokyo, Japan), and other chemicals from Wako Pure Chemical Industries Ltd. (Osaka, Japan). The MB49 bladder cancer cell line was provided by Jun Miyazaki (MD, PhD; International University of Health and Welfare, School of Medicine, Chiba, Japan).

### 4.2. Instruments

NIRF imaging was performed using a Photodynamic Eye-neo (PDE-neo) camera system (Hamamatsu Photonics K.K., Shizuoka, Japan) in the dark. The system was equipped with a 760-nm light-emitting diode for excitation, charge-coupled device for detection, and optical high-pass filter for efficient NIRF detection in front of the charge-coupled device detector. The measurement conditions were as follows: brightness: −2.5; contrast, 5; excitation level (Ex.): 0 (off) to 10 (maximum). The excitation level was adjusted arbitrarily. All images were obtained at the same distance between the mice and the camera.

The NIRF intensity was analyzed using the region-of-interest analysis program (Hamamatsu Photonics K.K.). The video images were recorded using a personal computer (Inspiron, Dell). The fluorescence intensity was analyzed using the ROI analysis program (Hamamatsu Photonics K.K.).

NIRF microscopic observation of tissue sections and MKN-45 cells was performed using an Axiovert 200 microscope (Carl Zeiss Co., Ltd., Oberkochen, Germany) equipped with an object lens Plan-Apochromat 20×/0.75 (Carl Zeiss Co., Ltd.) and a monochrome camera (Axio CamMRm; Carl Zeiss Co., Ltd.) at 20 °C. Microscopic NIRF was measured in the dark for 60 s in a 1 × 1 binning mode using the filter set 41037 Li-Cor for IR Dye 800 (excitation bandpass: 720–760 nm; emission long-pass: >780 nm; Chroma Technology, Bellows Falls, VT, USA), and images were obtained using AxioVision 4.8 software (Carl Zeiss Co., Ltd.).

Bladder tissues were stained with hematoxylin and eosin and observed under a microscope (ECLIPSE E600; Nikon Corporation, Tokyo, Japan) equipped with an object lens 20× (Nikon Corporation) and a camera (E8400; Nikon Corporation).

### 4.3. NIRF Imaging of the Cellular Uptake of ASP5354

MB49 cells were grown in a mixture of DMEM (100 mL), 200 mM glutamine (2 mL), penicillin–streptomycin (0.2 mL), and FBS (10 mL) at 37 °C in 5% CO_2_. A solution of MB49 cells (1 × 10^5^ cells/mL) was centrifuged (200× *g*, 20 °C, 5 min), and the obtained cells were incubated in 0.5 mL of a saline solution of 24-μM ASP5354 at 37 °C for 10 min. Next, the incubated solution was centrifuged (200× *g*, 20 °C, 5 min), and the obtained cells were washed five times by 0.5 mL of PBS; the centrifuged cells were suspended in 0.1 mL of PBS (pH 7.4). The cell suspension was then observed under a microscope.

### 4.4. Animal Study

NIRF imaging studies were performed at ITECHLAB Co., Ltd. (Gifu, Japan) and complied with the regulations for animal experiments published by the Institutional Animal Care and Use Committee of ITECHLAB Co., Ltd. (Registration No. ITL-22-MV-340). Female C57BL/6NCrl mice (age, 6 weeks; mean weight, 20 g) were purchased from Japan SLC, Inc. (Shizuoka, Japan). All mice were housed under specific pathogen-free conditions prior to experimentation. Experiments were performed after all mice were anesthetized using subcutaneous injections of ketamine (75 mg/kg) and medetomidine (1 mg/kg).

After anesthetizing the mice and removing the urine in the bladder, MB49 cell suspension (1 × 10^5^ cells/mL) was inoculated inside the urinary bladder of the mice (0.1 mL/body) and incubated for 2 h under anesthesia. MB49 cells were inoculated into the mouse bladder for 3–4 weeks. Bladder cancer growth was confirmed using a hematuria test.

### 4.5. NIRF in Normal Bladder via Intravesical Administration of ASP5354

After anesthetizing normal mice, urine in the bladder was removed and 2.4 or 24 μM ASP5354 (approximately 0.1 mL, *n* = 3 mice) was injected into the bladders. Five minutes after the injection, the inside of the bladder was rinsed five times with saline, following which the bladders were filled with saline. NIRF images from outside the bladders in vivo and inside the bladders after excision of the bladders were acquired using the PDE-neo camera system.

### 4.6. NIRF Imaging of Bladder Cancer via Intravesical Administration of ASP5354

Urine in the cancerous bladder was removed and 2.4 μM ASP5354 (approximately 0.1 mL, *n* = 3 mice) was injected into the cancerous bladders. One minute after intravesical administration, the inside of the bladder was rinsed five times with saline, after which the bladder was filled with saline, and NIRF images from outside of the bladder were acquired using a PDE-neo camera system. Subsequently, saline in the bladder was removed and 2.4 μM ASP5354 (approximately 0.1 mL) was injected into the bladder. After 4 min, the inside of the bladder was rinsed five times with saline, after which the bladder was filled with saline, and NIRF images from outside of the bladder were obtained. The bladder was removed and opened, and the NIRF image in the inner layer of the bladder was obtained.

Urine in the cancerous bladder was removed and 2.4 μM ASP5354 (approximately 0.1 mL) was injected into the bladder. After 5 min following the injection, the inside of the bladder was rinsed five times with saline and filled with saline, and NIRF images were obtained from the outside of the bladder. The bladder was removed and opened, and the NIRF image of the inner layer of the bladder was obtained. Subsequently, the bladder layer was sectioned and the NIRF image of the section was obtained.

### 4.7. Influence of Times of Bladder Rinsing for NIRF Imaging

Urine in the cancerous bladder was removed and 2.4 μM ASP5354 (approximately 0.1 mL, *n* = 3 mice) was injected into the bladder. After 5 min following the injection, the inside of the bladder was rinsed five times with saline and filled with saline, and NIRF images were obtained from the outside of the bladder. Subsequently, the inside of the bladder was rinsed five times with saline, and NIRF images from outside the bladder were obtained. In addition, the bladder was rinsed five times with saline and NIRF images from outside the bladder and in the inner layer of the bladder were obtained.

### 4.8. NIRF Imaging of Transition in the Bladder of ASP5354 after Intravenous Injection

ASP5354 was administered intravenously at 240 nmol/kg body weight to normal mice (*n* = 3 mice). After laparotomy, NIRF imaging of the internal organs was performed in the dark using a PDE-neo camera system at 10 min, 60 min, 5 h, 16 h, and 24 h after administration.

### 4.9. NIRF Imaging of Bladder Cancer via Intravenous Administration of ASP5354

ASP5354 was intravenously administered at a dose of 240 nmol/kg body weight. After 5 h (after rinsing five times with saline), 16 h (without rinsing), and 24 h (without rinsing), NIRF imaging from outside the bladder was performed (*n* > 3 mice for each waiting time). After completing in vivo NIRF imaging, bladder cancer was identified and NIRF images of the inner tissue were acquired. As a control, ASP5354 at 240 nmol/kg body weight was administered intravenously in mice with normal bladders, and NIRF imaging of the bladder was performed at 5 (after rinsing with saline), 16, and 24 h after ASP5354 injection (*n* = 3 mice for each waiting time).

### 4.10. Histopathological Analysis of Cancerous Bladder

After completing in vivo and ex vivo NIRF imaging of the cancerous bladder, bladder cancer tissues were excised. The tissue was frozen and sectioned at a thickness of 5 μm. Next, NIRF imaging of the sections was performed under a NIRF microscope. Another frozen section was stained with hematoxylin and eosin and observed under a microscope.

## 5. Conclusions

The ability of ASP5354 for real-time NIRF imaging of bladder cancer in a mouse model of MB49 bladder cancer was evaluated using an NIRF camera device. ASP5354 demonstrated the ability to be selectively uptaken in bladder cancer tissues following intravesical administration. Therefore, the NIRF emission clearly visualized the cancer tissue in the bladder. As ASP5354 has a remarkable accumulation in the bladder through specific and ultra-rapid renal clearance via intravenous administration, the intravenous administration method allows noninvasive real-time imaging of the bladder cancerous layer as intravesical administration. These new imaging techniques that use a combination of ASP5354 and NIRF imaging devices have the potential for real-time and sensitive detection of bladder cancer during clinical diagnosis and surgery.

## Figures and Tables

**Figure 1 ijms-24-02349-f001:**
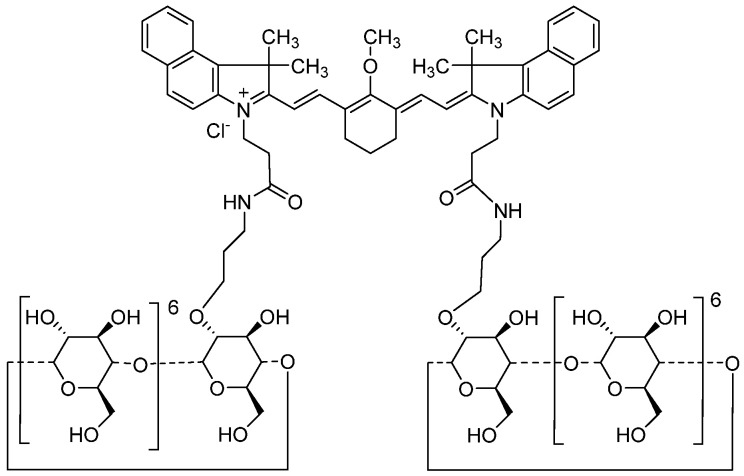
Chemical structure of ASP5354.

**Figure 2 ijms-24-02349-f002:**
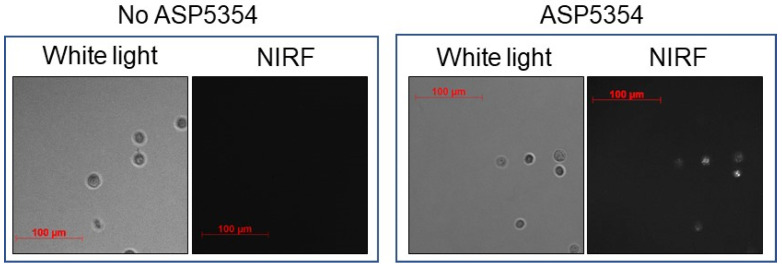
In vitro near-infrared fluorescence (NIRF) imaging of 24 μM ASP5354 uptake to mouse MB49 bladder cancer cells. NIRF images of cells with 60-s exposure after 10-min incubation NIRF is displayed in white.

**Figure 3 ijms-24-02349-f003:**
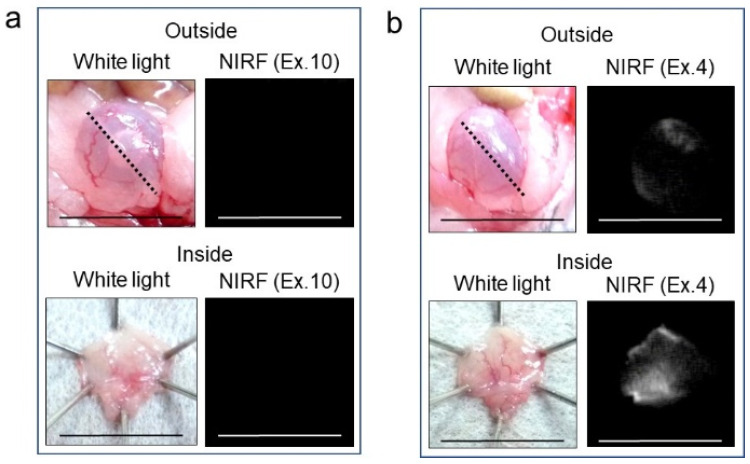
Near-infrared fluorescence (NIRF) imaging of six normal bladders. The 2.4-μM (**a**) or 24-μM (**b**) ASP5354 was injected in normal bladder, 5 min after which, the bladder was rinsed with saline 5 times, filled up with saline, and NIRF imaging was performed from outside the bladder. After imaging, the bladder was picked out and opened along the black dotted line. NIRF in the obtained bladder inner layer was imaged. Excitation levels (Ex.) are shown above the photos. NIRF is displayed in white. Scale bar represents 10 mm.

**Figure 4 ijms-24-02349-f004:**
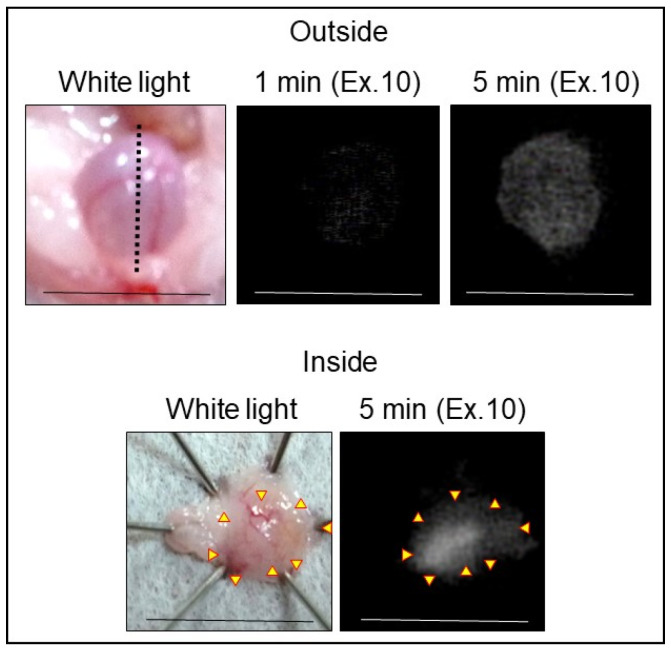
Representative near-infrared fluorescence (NIRF) imaging of MB49 bladder cancer with ASP5354. ASP5354 (2.4 μM) was injected in cancerous bladder, 1 or 5 min after which, the cancerous bladder was rinsed with saline 5 times, filled up with saline, and NIRF imaging was performed. After imaging for 5-min incubation, the bladder was picked out and opened along the black dotted line. NIRF in the obtained bladder inner layer was imaged. NIRF is displayed in white. Scale bar represents 10 mm. Yellow arrows show points of boundary between NIRF and non-NIRF.

**Figure 5 ijms-24-02349-f005:**
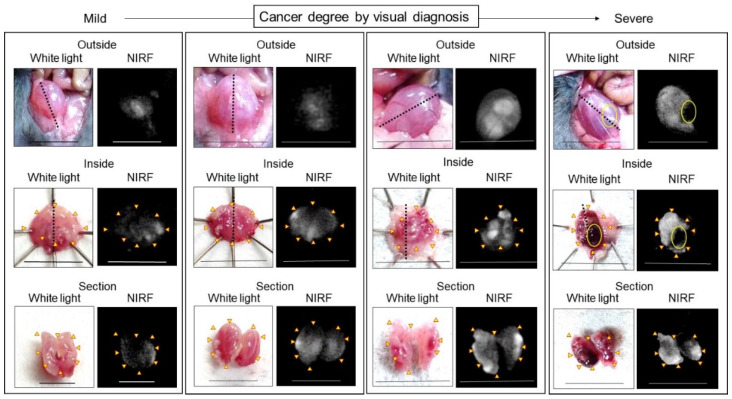
Representative near-infrared fluorescence (NIRF) imaging of MB49 bladder cancer in mice after intravesical administration of ASP5354. ASP5354 (2.4 μM) was injected in cancerous bladder, 5 min after which, the cancerous bladder was rinsed with saline 5 times, filled up with saline, and NIRF imaging was performed from outside the bladder. After imaging, the bladder was picked out and opened along the black dotted line. NIRF in the obtained bladder inner layer was imaged. All excitation levels are 10 a.u. NIRF is displayed in white. Scale bar represents 10 mm. Yellow arrows show points of boundary between NIRF and non-NIRF. Yellow circles in photos for Severe degree (Outside and Inside) show severe blood clot. Bladder (for Outside, Inside, and Section) in each frame is identical.

**Figure 6 ijms-24-02349-f006:**
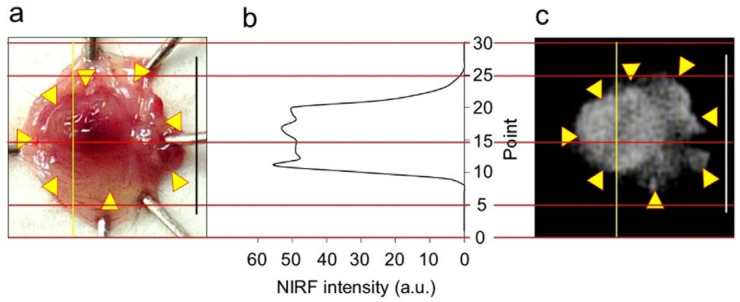
Near-infrared fluorescence (NIRF) intensity in cancerous bladder tissues. (**a**) Bladder inner layer under white light. (**b**) NIRF intensity on yellow line in (**a**,**c**). (**c**) NIRF image of bladder inner layer shown in photo (**a**). Scale bar in photos (**a**,**c**) represents 10 mm. Yellow arrows in photo (**a**) show points of boundary between NIRF and non-NIRF based on the photo (**c**). Red lines are scale lines common to (**a**–**c**).

**Figure 7 ijms-24-02349-f007:**
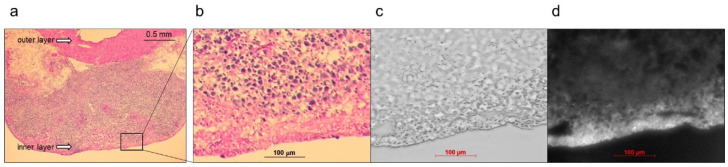
Representative near-infrared fluorescence (NIRF) images of cancerous bladder tissues in histopathological analysis. ASP5354 (2.4 μM) was injected in cancerous bladder, 5 min after which, the bladder was rinsed with saline 5 times, picked out, and opened. (**a**,**b**) Hematoxylin and eosin-stained tissues in the bladder. (**c**) Frozen section under white light. (**d**) NIRF of ASP5354 in frozen section (**c**). NIRF is displayed in white.

**Figure 8 ijms-24-02349-f008:**
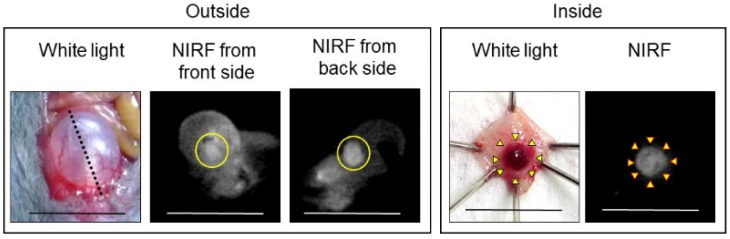
Representative near-infrared fluorescence (NIRF) imaging of MB49 bladder cancer after intravesical administration of ASP5354. ASP5354 (2.4 μM) was injected in cancerous bladder, 5 min post which, the cancerous bladder was rinsed with saline 5 times, filled up with saline, and NIRF images were obtained from front and back sides of the bladder. After imaging, the bladder was picked out, opened along the black dotted line, and NIRF in the obtained bladder inner layer was imaged. All excitation levels are 10 a.u. NIRF is displayed in white. Scale bar represents 10 mm. Bladder cancer is circled with yellow line. Yellow arrows show points of boundary between NIRF and non-NIRF.

**Figure 9 ijms-24-02349-f009:**
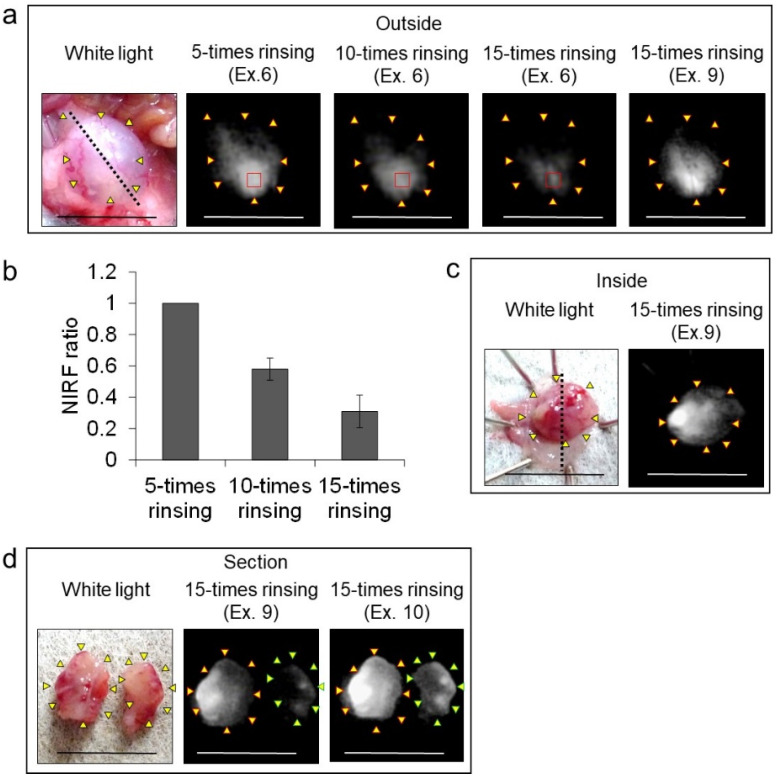
Effect of rinsing with saline on near-infrared fluorescence (NIRF) imaging of MB49 bladder cancer in three mice after intravesical administration of ASP5354. (**a**) Representative images of the three bladders. ASP5354 (2.4 μM) was injected in cancerous bladder, and 5 min after that, rinsed with saline 5, 10, or 15 times, filled up with saline, and NIRF imaging was performed from outside the bladder. Yellow arrows show points of boundary between NIRF and non-NIRF in photo for 5-times rinsing. (**b**) NIRF intensity ratio of the region of interest shown in red squares in photos (**a**). NIRF intensities are presented as means ± standard deviation (*n* = 3 bladders). (**c**) After the imaging from outside the bladder shown in photo (**a**), the bladder was picked out and opened along the black dotted line. NIRF in the obtained bladder inner layer was imaged. Yellow arrows show points of boundary between NIRF and non-NIRF. (**d**) The bladder inner layer shown in photo (**c**) was sectioned and NIRF image was obtained. Yellow and green arrows show points of boundary between NIRF and non-NIRF. Excitation levels (Ex.) are shown above photos. NIRF is displayed in white. Scale bar represents 10 mm.

**Figure 10 ijms-24-02349-f010:**
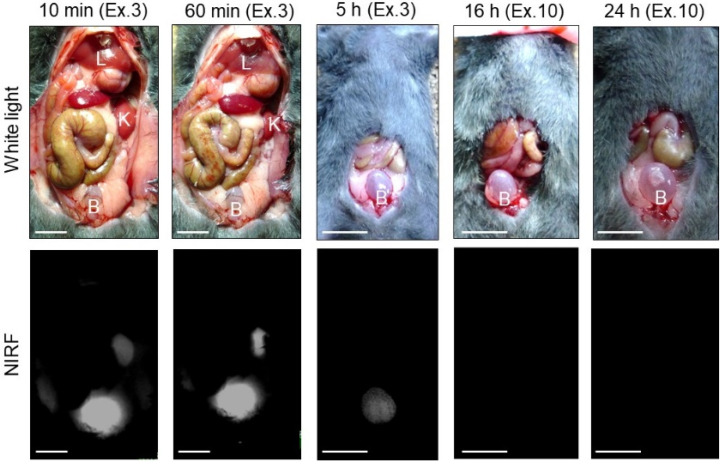
Representative near-infrared fluorescence (NIRF) imaging of normal bladders in three mice after intravenous administration of ASP5354 (240 nmol/kg body weight). Excitation levels (Ex.) are shown above photos. NIRF intensities in photos for 10 and 60 min are saturated because of too high intensity. NIRF is displayed in white. Scale bar represents 10 mm. B, bladder; K, kidney; L, liver.

**Figure 11 ijms-24-02349-f011:**
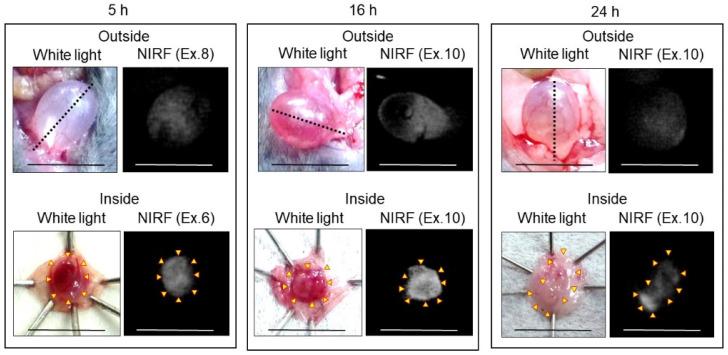
Representative near-infrared fluorescence (NIRF) imaging of MB49 cancerous bladders in three mice after intravenous administration of ASP5354. ASP5354 (240 nmol/kg body weight) was intravenously injected, 5 h (after rinsing with saline 5 times), 16 h (without rinsing), and 24 h (without rinsing) after injection, NIRF imaging was performed from outside the bladder. After imaging, the bladder was picked out and opened along the black dotted line. NIRF in the obtained bladder inner layer was imaged. Excitation levels (Ex.) are shown above photos. NIRF is displayed in white. Scale bar represents 10 mm. Yellow arrows show points of boundary between NIRF and non-NIRF.

## Data Availability

Data are contained within this article.

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
