# Peer review of "In Vivo Optical Imaging of Bladder Cancer Tissues in an MB49 Bladder Cancer Orthotopic Mouse Model Using the Intravesical or Intravenous Administration of Near-Infrared Fluorescence Probe"

_ijms, 2023, doi:10.3390/ijms24032349_

Round 1
Reviewer 1 Report
The authors explored a more efficient non-invasive optical imaging technique by proposing ASP5354, a near-infrared fluorescent probe that can be absorbed by bladder cancer. The authors found that ASP5354 provided non-invasive and specific in vivo optical imaging of MB49 bladder cancer by intravesical or intravenous injection of ASP5354. The authors believe this may provide a new avenue for the diagnosis and treatment of bladder cancer. At the same time, the authors believe that white light cystoscopy (WLC), also has some limitations, because it cannot grade tumor stage, determine invasion, or detect early-stage squamous carcinoma in CIS stage. However, in this paper, the authors did not study the advantages of ASP5354 in grading tumor staging and determining invasion. There needs to be more depth of analysis and clarification of these experiments to support the significance of the conclusions. This and the following issues need to be addressed.
1)In this paper, only one mouse bladder cancer cell line was used for validation, is it representative?
2)Figure 2: Lack of normal bladder cell uptake of ASP5354 as a control group.
3)Page 6, line 243. Spelling correction for ASP5354.
4)Does the picture (3a-c) represent a different meaning? Why do you have to put multiple pictures with the same meaning? The same problem also appears on several other pictures. Please make further revisions.
5)Figure 5: The intensity of NIRF released from severe cancer tissues is higher than that of mild cancer tissues as mentioned in the text. Please show the specific statistical method, was it observed by the naked eye?
6)In vivo optical imaging is one of the advantages of the NIR fluorescent probe ASP5354, why not detect it through the rat skin?
7)Whether there is any difference in NIRF imaging between non-muscle-invasive bladder cancer and muscle-invasive bladder cancer after injection of ASP5354, related experiments need to be improved.
Author Response
Response to Reviews
I carefully considered the comments from the reviewers.
Reviewer: 1
Comment 1:
The authors believe this may provide a new avenue for the diagnosis and treatment of bladder cancer. At the same time, the authors believe that white light cystoscopy (WLC), also has some limitations, because it cannot grade tumor stage, determine invasion, or detect early-stage squamous carcinoma in CIS stage. However, in this paper, the authors did not study the advantages of ASP5354 in grading tumor staging and determining invasion. There needs to be more depth of analysis and clarification of these experiments to support the significance of the conclusions. This and the following issues need to be addressed.
Response 1:
The normal thickness of the human bladder wall is approximately 15 mm and it becomes thinner as urine accumulates. The in vivo mouse bladder wall is thinner (<0.1 mm) than human bladder wall. In this study using mouse model, the cancer invasion into the deep bladder layer was uncontrolled after implantation due to thinness of bladder wall. Then, no experiments to evaluate difference in in vivo NIRF imaging between non-muscle-invasive bladder cancer and muscle-invasive bladder cancer using ASP5354 was successfully performed. This experimental limitation in this study using rat model is described in Discussion section, page 14, lines 476−489 (yellow highlight) in a new manuscript.
Comment 2:
In this paper, only one mouse bladder cancer cell line was used for validation, is it representative?
Response 2:
MB49 mouse bladder cancer cell is representative, because it has been extensively used for >40 years, as shown references 18 and 19.
Comment 3:
Figure 2: Lack of normal bladder cell uptake of ASP5354 as a control group.
Response 3:
In vitro ASP5354 uptake examination for normal bladder cell was not done because ASP5354 (24 mM) was up-taken normal bladder tissues also in five min, however not 2.4-mM ASP5354 in five min as shown Figure 3.
Comment 4:
Page 6, line 243. Spelling correction for ASP5354.
Response 4:
“ASP534” was corrected to “ASP5354” in a new manuscript.
Comment 5:
Does the picture (3a-c) represent a different meaning? Why do you have to put multiple pictures with the same meaning? The same problem also appears on several other pictures. Please make further revisions.
Response 5:
Pictures with the same meaning in Figures 3 and 4 were deleted in a new manuscript.
Comment 6:
Figure 5: The intensity of NIRF released from severe cancer tissues is higher than that of mild cancer tissues as mentioned in the text. Please show the specific statistical method, was it observed by the naked eye?
Response 6:
As shown in Figure 5, cancer tissue was visually diagnosed based on the red lesion and its width different from normal tissue under white light. This was described in page 6, line 264-265 (yellow highlight) in a new manuscript.
Comment 7:
In vivo optical imaging is one of the advantages of the NIR fluorescent probe ASP5354, why not detect it through the rat skin?
Response 7:
NIRF detection through the rat skin was not succeeded, because NIRF was not clearly detected due to its light diffusing.
Comment 8:
Whether there is any difference in NIRF imaging between non-muscle-invasive bladder cancer and muscle-invasive bladder cancer after injection of ASP5354, related experiments need to be improved.
Response 8:
Response 8 was described in Response 1.

Reviewer 2 Report
The research is well presented and description is very detailed. I would suggest to improve the presentation of the scale bar, if authors agree to do so. I would write:
"Scale bar represents 10 mm". Better formatation of images is suggested but not mandatory.
Congratulations on the very scientifically detailed work.
Author Response
Response to Reviews
I carefully considered the comments from the reviewers.
Reviewer: 2
Comment 1:"Scale bar represents 10 mm". Better formation of images is suggested but not mandatory.
Response 1:
In corrected manuscript, "Scale bar represents 10 mm" was used.

Round 2
Reviewer 1 Report
The authors explored a more efficient non-invasive optical imaging technique by proposing ASP5354, a near-infrared fluorescent probe that can be absorbed by bladder cancer. The authors found that ASP5354 provided non-invasive and specific in vivo optical imaging of MB49 bladder cancer by intravesical or intravenous injection of ASP5354. The authors believe this may provide a new avenue for the diagnosis and treatment of bladder cancer. At the same time, the authors believe that white light cystoscopy (WLC), also has some limitations, because it cannot grade tumor stage, determine invasion, or detect early-stage squamous carcinoma in CIS stage. However, in this paper, the authors did not study the advantages of ASP5354 in grading tumor staging and determining invasion. There needs to be more depth of analysis and clarification of these experiments to support the significance of the conclusions. This and the following issues need to be addressed.1) In this paper, only one mouse bladder cancer cell line was used for validation, is it representative?
2) Figure 2: Lack of normal bladder cell uptake of ASP5354 as a control group.
3) Page 6, line 243. Spelling correction for ASP5354.
4) Does the picture (3a-c) represent a different meaning? Why do you have to put multiple pictures with the same meaning? The same problem also appears on several other pictures. Please make further revisions.
5) Figure 5: The intensity of NIRF released from severe cancer tissues is higher than that of mild cancer tissues as mentioned in the text. Please show the specific statistical method, was it observed by the naked eye?
6) In vivo optical imaging is one of the advantages of the NIR fluorescent probe ASP5354, why not detect it through the rat skin?
7) Whether there is any difference in NIRF imaging between non-muscle-invasive bladder cancer and muscle-invasive bladder cancer after injection of ASP5354, related experiments need to be improved.
Author Response
Thank you so much for your reviewer 1 comments.
I read the referee report (round 2) of reviewer 1.
In the scores, "Are the conclusions supported by the results?" was changed from "Can be improved" in round 1 to "Yes" in round 2. Date was changed from "10 Jan" in the round 1 to "12 Jan" in the round 2. Other scores and reviewer's comments in the round 2 were the same as those in the round 1. I have already responded to referee's comments in the round 1 and corrected original manuscript in the round 1.
Round 3
Reviewer 1 Report
This manuscript involves a very interesting topic. ASP5354 is a more effective non-invasive optical imaging technology for diagnosing bladder cancer. It is also interesting because it may help early diagnosis of bladder cancer patients. This article is well written and well structured. The revised version can be considered for acceptance.